# The Simpson Grading: Is It Still Valid?

**DOI:** 10.3390/cancers14082007

**Published:** 2022-04-15

**Authors:** Silky Chotai, Theodore H. Schwartz

**Affiliations:** 1Department of Neurosurgery, Vanderbilt University Medical Center, Nashville, TN 37232, USA; silky.chotai@vumc.org; 2Department of Neurosurgery, Otolaryngology and Neuroscience, Weill Cornell Medicine, New York-Presbyterian Hospital, New York, NY 10065, USA

**Keywords:** meningioma, resection, Simpson, grades, intraoperative, imaging

## Abstract

**Simple Summary:**

Since the initial publication of the Simpson Grade scale, the management paradigm for intracranial meningiomas has significantly evolved. The generalized application of the Simpson Grade in modern neurosurgery management of meningiomas is controversial. We conducted a review of the literature to determine the prognostic significance of the Simpson Grade and find it to be an antiquated grading scale with limited utility in the modern era.

**Abstract:**

The Simpson Grade was introduced in the era of limited resources, outdated techniques, and rudimentary surgical and imaging technologies. With the advent of modern techniques including pre- and post-operative imaging, microsurgical and endoscopic techniques, advanced histopathology and molecular analysis and adjuvant radiotherapy, the utility of the Simpson Grade scale for prognostication of recurrence after meningioma resection has become less useful. While the extent of resection remains an important factor in reducing recurrence, a subjective naked-eye criteria to Grade extent of resection cannot be generalized to all meningiomas regardless of their location or biology. Achieving the highest Simpson Grade resection should not always be the goal of surgery. It is prudent to take advantage of all the tools in the neurosurgeons’ armamentarium to aim for maximal safe resection of meningiomas. The primary goal of this study was to review the literature highlighting the Simpson Grade and its association with recurrence in modern meningioma practice. A PubMed search was conducted using terms “Simpson”, “Grade”, “meningioma”, “recurrence”, “gross total resection”, “extent of resection” “human”. A separate search using the terms “intraoperative imaging”, “intraoperative MRI” and “meningioma” were conducted. All studies reporting prognostic value of Simpson Grades were retrospective in nature. Simpson Grade I, II and III can be defined as gross total resection and were associated with lower recurrence compared to Simpson Grade IV or subtotal resection. The volume of residual tumor, a factor not considered in the Simpson Grade, is also a useful predictor of recurrence. Subtotal resection followed by stereotactic radiosurgery has similar recurrence-free survival as gross total resection. In current modern meningioma surgery, the Simpson Grade is no longer relevant and should be replaced with a grading scale that relies on post-operative MRI imaging that assess GTR versus STR and then divides STR into > or <4–5 cm^3^, in combination with modern molecular-based techniques for recurrence risk stratification.

## 1. Introduction

Meningiomas are one of the most common primary brain tumors comprising approximately one-third of all intracranial tumors [1,2,3]. For symptomatic meningiomas, surgical resection with the aim of achieving complete safe resection remains the mainstay of treatment. In 1957, Donald Simpson published a grading scale to stratify the extent of resection of intracranial meningiomas based on subjective naked eye, intraoperative observation, which he associated with recurrence rates, determined by symptomatic progression [4]. Since then, Simpson grading has been widely used to categorize the extent of resection as a predictor of meningioma recurrence. Since its publication, the 5-point Simpson grading system has garnered much attention and engendered even more controversy [5,6,7,8,9,10,11,12,13,14,15,16]. Questions have been raised regarding its subjectivity, generalizability to all intracranial meningiomas regardless of location and its prognostic value. In recent years, with advent of preoperative, intraoperative, and postoperative imaging techniques, novel molecular classifications and robust microscopic, endoscopic, and other intraoperative visualization adjuncts, the value of Simpson Grade scale in predicting meningioma recurrence is diminishing. To that end, a critical analysis of published studies in recent years is needed. This review will highlight the limitations and benefits of Simpson grading system and discuss its utility in modern meningioma practice. 

## 2. Materials and Methods

An advanced PubMed search was conducted using combinations of the following key search terms: “Simpson”, “Grade”, “meningioma”, “recurrence”, “gross total resection”, “extent of resection” and “human”. Only English language articles were reviewed. A separate search using the terms “intraoperative imaging”, “intraoperative MRI” and “meningioma” were conducted. The primary goal of this study was to review literature highlighting Simpson Grade resection and its association with recurrence in the modern meningioma practice. The first search yielded 31 studies. Of those, studies focusing on spinal meningioma only were not reviewed further. If a study reported mixed location of meningioma and spine was one of the locations that study was reviewed further. The second search term yielded 10 studies. Other studies were identified by cross-referencing. A narrative review of these studies is summarized. 

## 3. Results

### 3.1. Analysis of Original Manuscript

In 1957, Donald Simpson published a retrospective series of 332 intracranial meningiomas operated on at two centers—242 from Simpson’s series and 97 from Cairn’s series. These patients were operated between 1928 and 1954. Studies focusing on spinal and intraorbital meningiomas were excluded. The primary aim of the article was to report the frequency and factors associated with recurrence [4]. 

In their analysis, Simpson noted that the scope of resection was not same for all meningiomas. It varied based on their location and invasion of surrounding structures such as brain parenchyma, dura, venous sinuses, and bone. Based on his experience, he quantified the extent of tumor resection into 5 Grades (Table 1). He noted that about 90 patients had Grade I resection and of those 8 recurred (9%); a Grade II resection was achieved in 114 patients, and of those 18 recurred (19%); in those with incomplete resection Grades III–V–the incidence of recurrence was higher. This publication was received with great enthusiasm. Since then, the Simpson Grade has been used to quantify the extent of meningioma resection and predict recurrence-free survival. However, in recent years, several studies have questioned the applicability and generalizability of Simpson grading [12,13,15,17]. Schwartz et al. summarized the limitations of Simpson Grade in their recent review paper [13]. 

There are several limitations of the original study. First, the study was designed as a retrospective case series. Although, in aggregate the sample size of 332 is large, the number of patients in each subgroup of meningioma based on location was inadequate to derive any meaningful conclusions. Furthermore, the diagnosis of meningioma was based on limited imaging and histopathology availability at that time. Meningioma behavior is now understood to be better defined by World Health Organization (WHO) Grades [18], tumor proliferation index [19], tumor invasion [20,21] and molecular subtypes [22]. It is possible that if the meningiomas from the original article were reclassified based on the modern diagnostic methods, tumor pathology might have higher predictive significance than subjective naked-eye assessments of extent of resection. Regarding extent of resection, as noted by Simpson, these were determined by retrospective chart review. It is possible that minor operative details might be missed, and the final extent of resection reported is at the mercy of the interpreter of the note. Furthermore, the clinical utility of a resection Grade based on a subjective intraoperative assessment by the surgeon may be questionable when more objective predictors of recurrence are now available. 

Interestingly, given the absence of postoperative imaging, recurrence was defined based on the reappearance of symptoms that could be attributed to tumor growth after a period of symptomatic relief. Therefore, the recurrence rate reported in the study might be an underestimation of the actual recurrence rate. 

### 3.2. Simpson Grade 0 

The recurrence rate after achieving a Simpson Grade I resection has been reported in the range of 10% to 32% within 10 years postoperative follow-up [18,23,24]. This has been historically attributed to a limited extent of resection including the extent of surrounding dura resected. More than three decades after publication of Simpson Grades, Borovich and Doron demonstrated that more than 1/3rd of meningiomas demonstrate regional multifocality, with the existence of small macroscopic and microscopic tumor cells within the dura as far as 4 cm from the edge of the main tumor [25,26]. Soon after that, Kino et al. reported their experience of 37 supratentorial convexity meningiomas with removal of additional dural margin of about 2 cm around the tumor and labelled this a Grade 0 resection [9]. At 1 to 10 years clinical and radiological follow-up, no recurrences were noted in their series. The authors concluded that the recurrence rate for convexity meningiomas can be decreased by including in the resection a 2 cm margin of dura that might harbor a foci of tumor cells [9]. Morokoff et al. in a retrospective series of convexity meningiomas, reported a 1.8% 5-year recurrence rate after a 5-mm margin of surrounding dural resection [27]. 

Another unanswered question regarding the optimal extent of resection concerns the etiology of the dural tail and its oncologic potential. Several authors have reported the histopathological characteristics of the enhancing dural tail [28,29,30,31]. In a retrospective study of 179 convexity meningiomas, Qi et al. reported that the appearance of dural tail on preoperative MRI was significantly different in WHO Grade I vs. non-Grade I meningiomas [23]. The authors noted that dural tail with no nodularity were seen only in WHO Grade I meningiomas, while only some Grade I and all non-WHO Grade I meningioma demonstrated nodularity in the dural tail. The authors noted that the extent of tumor invasion in dura was <1.5 cm for smooth dural tails, whereas it was up to 2.5 cm for nodular dural tails. Based on these the authors recommended at least 2.5 m of dura should be resected when feasible and in cases where that extent of resection cannot be achieved the type of dural tail on preoperative MRI could be used to tailor extent of dural resection. 

While defining the optimal extent of dural resection or coagulation remains a topic of debate, several studies have reported tools to better define dural invasion intraoperative [28,29,32,33,34,35]. Raman spectroscopy has been advocated to define dural invasion intraoperatively to determine the extent of meningioma invasion into the dura, thereby facilitating resection of the involved dura [34,35]. Meningiomas have a high expression of somatostatin receptor 2. Other studies have demonstrated the use of somatostatin receptor 2A labeled fluorescence to determine dural invasion [32,33].

### 3.3. Simpson Grades I–III 

For the Simpson Grades I, II and III, the distinction between Grades I and II might be challenging when the Grades are derived based on review of intraoperative notes, or surgeons’ subjective observations of dural management. Furthermore, Grade I resection is often not possible for certain location such as skull base meningiomas, or meningiomas involving dural venous sinuses. Table 2 summarizes the studies reporting the Simpson grading and its relationship with recurrence. 

Several retrospective studies have demonstrated no differences in the recurrence rates between Simpson Grades I, II and III [10,17,36,37]. Particularly when a surgical microscope is employed, Simpson Grade I, II and III can all be considered as a gross total resection, with little difference between grades [17]. Figure 1 demonstrates a bar diagram showing recurrence-free survival probabilities based on Simpson Grade as reported in the studies reviewed. In a recent retrospective review of 939 meningiomas, Brokinkel et al. utilized time-dependent receiver operator curve analysis to compare recurrence following Grades I–III vs. Grades I–II resection and demonstrated that gross total resection defined as Grades I–III allowed more precise prediction of risk of recurrence than those defined as only Simpson Grades I–II. This suggests that from statistical standpoint, radical meningioma resection can be defined more simply as total removal of tumor with or without dural resection or coagulation [6]. In a separate study using the same data, Spille DC et al. investigated the prognostic value of Simpson grading. The authors noted that the extent of resection according to Simpson Grade was overrated in about 8% of meningiomas. There was residual tumor on postoperative MRI for resection graded as Simpson I or II. This underscores the subjectivity of Simpson grading compared to the more objective technique that relies on post-operative contrast-enhanced MRI scans. In a multivariable analysis, the higher histology grade and postoperative tumor volume were predictors of recurrence. Simpson Grade was not identified as predictor of recurrence [14].

VoB et al. emphasized the value of meningioma location in predicting recurrence. The authors noted that in 268 convexity meningiomas, the frequency of tumor recurrence was higher for Simpson Grades III and IV, but there was no difference between Grades I and II. In 325 skull-base meningiomas, the risk of recurrence increased only after Grade IV resection [15]. In a study by Shugrue et al., specifically concerning skull base meningiomas, recurrences were not fewer for lower Simpson Grade resections [17]. Similarly, in a retrospective series of convexity and skull base meningiomas, Heald et al. reported that the recurrence was higher in patients with Simpson Grade IV compared with Grade I or II resections; however, there was no difference in recurrence rates between those who had Simpson Grade I vs. II resections. Similar results with no difference in recurrence-free survival between Grades I and II have been reported by other authors [17,38]. In a retrospective analysis of 1571 meningioma, Behling et al. demonstrated that in a multivariable analysis, there was no prognostic effect of dural resection compared to coagulation, while dural coagulation had benefit compared to leaving the dural attachment untreated. The authors suggested that optimal coagulation of dura might be as effective as radical resection of dural attachment [5]. This suggests that a safe gross total resection should be attempted, however a heroic tumor resection with extended dural resection might not be beneficial in providing additional recurrence-free survival [39]. 

On the other hand, a few authors have emphasized the superiority of Simpson Grade I resections [11,14,40,41]. Przybylowski et al. in a retrospective case series of 492 patients with WHO Grade I convexity and skull base meningioma, noted that the Simpson Grade I resection resulted in superior recurrence free survival compared to Simpson Grade II resection [11]. Hasseleid et al. compared Simpson Grade I resection with Grade II + III and demonstrated that the those with Grades II + III had 4.9 times higher chances of recurrence compared to Grade I, after adjusting for WHO grading. This suggests that if complete resection of tumor with dural resection is not achieved the chances of recurrence increases [40]. Nanda et al. in retrospective series of 458 patients with WHO Grade I meningioma reported that the recurrence rates for Simpson Grades I, II and III were 5%, 22% and 31%, respectively [41]. The authors noted that Simpson Grade III resection was 13.1 times more likely to recur than a Simpson Grade I resection. 

Alvernia JE et al. in a retrospective review of 100 patients with convexity meningioma demonstrated that meningiomas with pial involvement or vascular attachments might behave differently [42]. They proposed a modified Simpson Grade III resection to divide into grade IIIa and IIIB. The grade IIIa was defined as a small layer of tissue left at the cortex due to adherence to pia and is only visible under microscope and grade IIIB was defined as a small visible tumor left because of invasion of cortical vessel in eloquent area which is coagulated under the microscope. Of nine cases with grade III resection (three with grade IIIa, six with grade IIIb), there were 0 recurrences in grade IIIa and two recurrence (22.2%) in grade IIIb subgroup. Overall, the number of patients with non-Simpson Grade I resection were fewer. Of note, recurrence rate with Simpson Grade I resection was 2.2%, which is higher than with grade IIIa resection.

**Table 2 cancers-14-02007-t002:** Summary of recent studies reporting the Simpson Grade and its association with recurrence after meningioma resection.

Author/Year/Study Design	No of Patients	Simpson Grade (SG)	RFS	Median/Mean Follow-Up Months	WHO Grades	Location	SG Associated with Recurrence
Sughrue et al. [17] 2010 Retrospective	373	SG I: 88, SG II: 114, SG III: 57, SG IV: 114	(5-yr) SG I: 95%, SG II: 85%, SG III: 88%, SG IV: 81%	44.4 (median) (6 m–18 yrs)	I	Convexity, Skull base, parasagittal	No
Alvernia et al. [42] Retrospective	100	SG I: 91 SG II: 0 SG III: 9	RFS NR Recurrence rate SG I: 2.2% SG IIIa: 0 SG IIIb: 22%	86 m (median) (2–16 yrs)	I, II	Convexity	Yes
Oya et al. [37] 2012 Retrospective	240	SG I: 63, SG II: 104, SG III: 35, SG IV: 43	(5-yr) SG I: 97.6%, SG II: 87.7%, SG III: 84.1%, SG IV: 56.8%	NR	I	Convexity, Skull base, parasagittal	SG IV: shorter RFS No difference in RFS between SG: I–III
Hasseleid et al. [40] 2012 Retrospective	391	SG I: 315, SG II: 46, SG III: 16, SG4 IV: 12,	Overall: SG I: 96.8%, SG II: 84.8%, SG III: 87.5%, SG IV: 50%	85.2 (median) 45.6 m (6 m–108 m)	I, II, III	Convexity, excluded tumor involving sagittal sinus	Simpson II + III and IV + V had higher recurrence than Grade I
Heald et al. [39] 2014 Retrospective	183	SG I: 71, SG II: 74, SG III: 0, SG IV: 33	(3-yr) SG I: 95%, SG II: 87%, SG III: NA, SG IV: 67%	35.3 (mean) (6 m–81.6 m)	I	Convexity, Skull base, parasagittal	Yes
Otero-Rodriguez et al. [10] 2016 Retrospective	224	SG I: 54, SG II: 86, SG III: 84	(5-yr) SG I: 97%, SG II: 95%, SG III: 98%,	60 (median) (NR)	I	Convexity, Skull base, parasagittal	No difference in recurrence rates between SG I–III
Gousias et al. [43] 2016 Retrospective	901	SG I: 570, SG II: 197, SG III: 92, SG IV: 35	(10-yr) SG I: 91.8%, SG II: 81.2%, SG III: 71.8%, SG IV: 65.3%	62 (median) (NR)	I, II, III	Convexity, Skull base, parasagittal	Yes
Nanda et al. [41] 2017 Retrospective	458	SG I: 80, SG II: 294, SG III: 32, SG IV: 52	Overall SG I: 95%, SG II: 78%, SG III: 69%, SG IV: 65%	54 (mean) (1 m–250 m)	I	Convexity, Skull base	Yes
Winther et al. [16] 2017 Retrospective	113	SG I: 35, SG II: 48, SG III: 16, SG IV: 14	(5-yr) SG I: 97.1%, SG II: 91.3%, SG III: 86.7%, SG IV: 54.5%	123 (median) (6.9 m–210.6 m)	I	Convexity, Skull base, parasagittal	Yes
Ehresman et al. [36] 2018 Retrospective	572	SG I: 125, SG II: 197, SG III: 92, SG IV: 158	(4-yr) SG I: 90.7%, SG II: 88.9%, SG III: 83.8%, SG IV: 72.7%	53.9 m (median) (24 m–83.9 m)	I, II, III	Convexity, Skull base, parasagittal	No. No difference between SG I and II.
VoB KM et al. [15] 2017 Retrospective	826	SG I: 238, SG II: 343, SG III: 102, SG IV: 79	SG I: 90.7%, SG II: 88.9%, SG III: 83.8%, SG IV: 72.7%	50 m (median) (0–277 m)	I, II, III	Convexity, Skull base, parasagittal	No difference in recurrence between I, II and III, increased risk after IV.
Przybylowski et al. [11] 2020 Retrospective	492	SG I: 97, SG II: 142, SG III: 50, SG IV: 152	(5-yr) SG I: 94.6%, SG II: 88.3%, SG III: 85.1%, SG IV: 55.6%, SG IV with radiosurgery: 85%	44.8 (mean) (SD:30.5)	I	Convexity, Skull base, parasagittal	Yes
Brokinkel et al. [6] 2020 Retrospective	939	SG I: 280, SG II: 446, SG III: 103, SG IV: 106	SG I: 92%, SG II: 89%, SG III: 82%, SG IV: 81%	37 m (median) (0–284 m)	I, II, III	Convexity, Skull base, parasagittal	Yes, the predictive value of SG is higher when dichotomizing into Grades I–III compared to I–II.
Behling et al. [5] 2021 Retrospective	1571	SG I: 376, SG II: 408, SG III: 303, SG IV: 484	SG I: 83.8%, SG II: 91.7%, SG III: 81.2%, SG IV: 59.1%	38.4 (mean) (1.2 m–195.6 m)	I, II, III	Convexity, Skull base, Parasagittal, Spinal	No
Spille D et al. [14] 2021 Retrospective	939	SG I: 280, SG II: 446, SG III: 103, SG IV: 106	SG I: 92%, SG II: 89%, SG III: 82%, SG IV: 81%	37 m (median) (NR)	I, II, III	Convexity, Skull base	Yes. Postoperative tumor volume predicts the risk of recurrence more relevantly than the Simpson Grade

NR = not reported, SG = Simpson Grade.

### 3.4. Simpson Grade IV–V

For higher Simpson Grade, as described in Simpson’s original paper, the distinction between Grade IV and V is subjective. Unless the size of residual tumor and proportion of resected tumor are quantified, a subtotal resection and limited decompression could be graded equally. Furthermore, the size and extent of residual tumor can impact the postoperative course, as well as the need for stereotactic radiosurgery. Given these limitations, Grades IV and V are commonly cited in the literature as one entity—subtotal tumor resection [6,7,37,44]. 

A few studies have demonstrated that the extent of residual tumor following subtotal resection is associated with recurrence-free survival. In a retrospective series of 65 patients who underwent Simpson Grade IV resection, Fukushima et al. demonstrated that postresection tumor volume of 4 cm^3^ or more was associated with a higher recurrence rate [45]. Similarly, Martini et al. demonstrated that among patients with Grade IV resection, the median growth rate was 0.09 cm^3^/year and residual tumor volume >5 cm^3^ was associated with higher absolute growth rates [46]. Thus, when gross total resection of tumor cannot be achieved the goal should be to minimize the volume of residual tumor such that it is amenable to postoperative stereotactic radiosurgery. Some authors have even demonstrated that the combination of radiotherapy with subtotal resection is associated with similar recurrence-free and overall survival rates as gross total resection [7,8,11]. Przybylowski et al. noted that Simpson Grade II and III resection had lower recurrence free survival compared to Simpson Grade IV resection without adjuvant radiosurgery but had similar recurrence free survival compared to Simpson Grade IV resection with adjuvant radiosurgery [11]. 

### 3.5. Simpson Grade and WHO Grade

Of total 14 studies, seven reported recurrences based on WHO grading [5,6,14,15,36,40,43]. The higher WHO grade was a significant predictor of higher recurrence rates and lower recurrence free survival rate. In context of extent of resection and WHO grading, VoB KM et al. demonstrated that in age, sex and WHO-grade adjusted analysis the risk of recurrence increased only after Simpson grade IV resection or STR for convexity and skull base meningiomas [15].

### 3.6. Intraoperative Imaging 

The gross total resection rates have improved with the introduction of microsurgical techniques, increased use of endoscopic techniques, use of 5-aminolevulinic acid, and use of ultrasound and intraoperative MRI (iMRI) [47,48,49,50,51,52]. Few studies have reported role of iMRI on extent of resection for meningioma [48,50,53]. Table 3 summarizes the studies reporting iMRI for meningioma resection. In a retrospective series of 27 patients with complex skull base meningiomas operated on by using iMRI assistance, Soleman et al. reported that utility of iMRI for meningioma surgery is limited. They noted that additional resection was carried out in only one patient based on information from iMRI; and that additional resection did not change the Simpson Grade [54]. 

In a prospective case series of 19 parasellar meningiomas operated on with iMRI guidance, Giodrano et al. noted that the benefit of iMRI for patients undergoing partial resection cannot be translated to Simpson Grade, as it remains Grade IV. However, the use of iMRI allowed for increasing extent of safe resection in 56% of cases and offered a better precondition for radiotherapy specifically for cavernous sinus and recurrent meningiomas [44]. Terpolilli et al. reported that the use of intraoperative CT for the surgery for 19 orbital meningiomas was useful in evaluating the residual osseous part of the lesion in 52% of the cases, and thereby allowing for sufficient decompression of the optic nerve resulting in improved outcomes. The authors did not quantify extent of resection and its association with recurrence [47]. In a recent case series of six patients with meningiomas located in eloquent areas or major dural sinuses, the use of iMRI was found to be useful in deciding additional tumor resection and to evaluate residual tumor volume [52]. None of these studies have demonstrated if iMRI had added benefit with dural resection or coagulation, emphasizing that intraoperative adjuncts might not have any added benefit once a Grade 3 resection is achieved [51]. Few authors have demonstrated that use of intraoperative 5-ALA guidance and fluorescein may aid in identification of tumor remnants and hyperostotic bone but the detection of dural infiltrations remains a matter of further investigation [49,55,56,57].

## 4. Discussion

Although the Simpson Grade, as originally conceived in 1957, may be antiquated, extent of resection remains a good predictor of recurrence for meningiomas. In 1957, when the meningioma biology was unclear, aggressive measures with dural resection were justifiable. With numerous revolutions in surgical technique, imaging, histopathology, molecular biology, and stereotactic radiosurgery the risk associated with the heroic measures required to achieve Grade 0 or Grade I in all tumors, must be balanced with the ability to achieve a similar outcome using other adjuncts, such as radiosurgery. Rather than relying on an intraoperative assessment of surgical accomplishments, prognosis should be determined based on the presence of residual tumor and its volume on post-operative contrast enhanced MRI, meninigioma location, WHO grading, histopathological and molecular subtype [6,14,18,19,20,21,22,23,24,58,59,60,61,62].

The goal of surgery should be the maximal safe resection of tumor with resection or coagulation of 1–2 cm of surrounding dura when possible. In three recent studies with larger study cohort [6,14,15], Simpson grading scale itself did not predict recurrence-free survival. The distinction between Simpson Grades I–III compared with Simpson Grade IV and V is the strongest predictor. Furthermore, a recent study analyzed Simpson Grade in context of WHO grading and demonstrated that after adjusting for WHO grade the risk of recurrence increases only after Simpson grade IV resection or STR regardless of meningioma location. This suggests that a two-scale model with either GTR or STR may be more than adequate, combined with molecular characteristics [5]. Studies focusing on factors predicting meningioma recurrence after subtotal resection suggests that minimizing the residual tumor volume less than 4–5 cm^3^ might be associated with higher recurrence free survival. The goal here should be minimize the residual tumor volume such that it is amenable to postoperative stereotactic radiosurgery. 

The iMRI has limited role in meningioma surgery [54]. For complex skull base tumors, using iMRI might assist in increasing extent of resection and decompressing the surrounding neurovascular structures [47,48,52]. However, its utility in improving Simpson grading is limited. The development of intraoperative tools such as 5-ALA, Raman spectroscopy, and somatostatin receptor 2A labeled fluorescence might be able to assess the dural infiltration with a high sensitivity thereby facilitating resection of the involved dura if needed [29,30,32,33,35]. 

## 5. Conclusions

In current modern meningioma surgery practice, the Simpson grade is no longer relevant and should be replaced with a grading scale that relies on post-operative MRI imaging that assess GTR versus STR and then divides STR into > or <4–5 cm^3^, in combination with modern molecular-based stratification for risk of recurrence. 

## Figures and Tables

**Figure 1 cancers-14-02007-f001:**
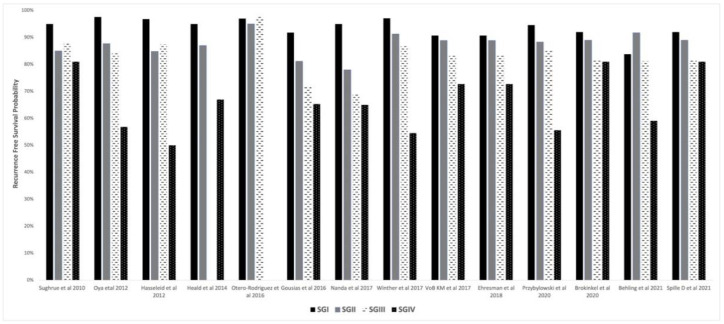
Bar Diagram demonstrating Recurrence-Free Survival Probabilities based on Simpson Grade reported in the literature.

**Table 1 cancers-14-02007-t001:** The Simpson Grade.

Grade	Definition	Number of Patients Treated	Recurrence as Reported
I	Macroscopic complete tumor resection with removal of affected dura and bone, when tumor arises from wall of dural venous sinus such an operation necessities resection of the sinus	90	(9%)
II	Macroscopic complete tumor resection and of its visible extension with coagulation of affected dura	114	18 (19%)
III	Macroscopic complete tumor removal	24	8 (29%)
IV	Partial resection, leaving intradural tumor in situ	51	20 (39%)
V	Decompression with/without biopsy	9	8 (88.9%)

Derived from the information provided in [4] Simpson D: The recurrence of intracranial meningiomas after surgical treatment.

**Table 3 cancers-14-02007-t003:** Summary of studies reporting utility of intraoperative imaging to enhance extent of resection for meningioma.

Author, Year, Study Design	Number of Tumors Treated	Location	iMRI/iCT Scan Utility	Impact of Intraoperative Imaging on Simpson Grade
Giordano et al. [44] 2019 Prospective	19	Parasellar	iMRI allowed the further safe resection in 56% of cases and offered a better precondition for radiotherapy. Increased EOR for 2/5 tuberculum sellae meningioma, an 5/9 cavernous sinus meningioma.	No change in Simpson Grade
Multani et al. [50] 2020 Retrospective	11	NA	5/11 (45.5%) iMRI detected residue and 3/5 additional resection was achieved	No mention of Simpson Grades
Ashour R et al. [53] 2016 Retrospective	10	Skull base	Additional resection in 4 meningiomas	No mention of Simpson Grades
Terpolilli et al. [47] 2016 Retrospective	19	Orbital meningioma	Intraoperative CT was used to evaluate the residual osseus part and therefore allowed for sufficient decompression of optic nerve in 52% of cases.	No mention of Simpson Grades
Soleman et al. [54] 2012 Retrospective	27	Skull base	Only one patient (3.4%) underwent resection of tumor remnant after iMRI, although without improvement of the Simpson resection Grade.	No change in Simpson Grade
Schulder et al. [48] 2001 Retrospective	4	Skull base	Amount of residual tumor was optimized for SRS	No change in Simpson Grade
Tuleasca C et al. [52] 2021 Case series	6	Eloquent areas, or dural sinus	Useful to increase EOR and reduce residual volume	GTR achieved after iMRI use in at least 2/6 patients. No change in EOR for 1 patient, other details not reported

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
