# Peer review of "The Simpson Grading: Is It Still Valid?"

_cancers, 2022, doi:10.3390/cancers14082007_

Round 1

Reviewer 1 Report

The authors present a review article on the prognostic value of the Simpson-scale in modern meningioma surgery. The article is interesting and well written. However, it mainly confirms the recent work of Schwartz et al. (which is correctly citied in the paper [13]). Novel aspects (compared to the review of Schwartz et al.) include the discussion of intraoperative imaging techniques suggested to improve the extend of surgical resection. 

Author Response

Reviewer 1:

The authors present a review article on the prognostic value of the Simpson-scale in modern meningioma surgery. The article is interesting and well written. However, it mainly confirms the recent work of Schwartz et al. (which is correctly citied in the paper [13]). Novel aspects (compared to the review of Schwartz et al.) include the discussion of intraoperative imaging techniques suggested to improve the extend of surgical resection.

Response:  Thank you for the comments.

Reviewer 2 Report

Chotai and Schwartz made a comprehensive review on the Simpson Grading. The authors conclude that the Simpson Grading is outdated considering the advance of many modern techniques including pre and post-operative imaging, microsurgical and endoscopic, advanced histopathology and molecular analysis and adjuvant radiotherapy. In general I agree with the authors and recommend publication of this manuscript if the following points can be addressed.

  1. The manuscript has three tables but no figures. I think adding some figures will greatly improve the manuscript.
  2. The main argument for outdating Simpson Grade, which is a product of limited resources and old techniques, is the development of modern techniques. However, it still seems unclear to me how modern techniques perform better. A statistical comparison would be helpful. In addition, how about the cost of switching from the old-technique-based Simpson Grade to something new that is based on modern techniques? How is the availability of these modern techniques in developing countries?

Author Response

Reviewer 2:
Chotai and Schwartz made a comprehensive review on the Simpson Grading. The authors conclude that the Simpson Grading is outdated considering the advance of many modern techniques including pre and post-operative imaging, microsurgical and endoscopic, advanced histopathology and molecular analysis and adjuvant radiotherapy. In general, I agree with the authors and recommend publication of this manuscript if the following points can be addressed.

  1. The manuscript has three tables but no figures. I think adding some figures will greatly improve the manuscript.

Response: Thank you for the comments. Figure 1 is added which demonstrates a bar diagram showing recurrence-free survival probabilities based on Simpson Grade as reported in the studies reviewed.

2. The main argument for outdating Simpson Grade, which is a product of limited resources and old techniques, is the development of modern techniques. However, it still seems unclear to me how modern techniques perform better. A statistical comparison would be helpful. In addition, how about the cost of switching from the old-technique-based Simpson Grade to something new that is based on modern techniques? How is the availability of these modern techniques in developing countries?

Response: As discussed in result section and discussion the intraoperative MRI has limited role in meningioma surgery. To that end, we do not suggest switching to costly techniques to perform better. The goal of surgery should be the maximal safe resection tumor with resection or coagulation of 1-2 cm of surrounding dura when possible. This suggests that a two-scale model with either GTR or STR may be more than adequate, combined with molecular characteristics. While the detailed molecular-based stratification for risk of recurrence may not be feasible in developing countries, a postoperative MRI imaging-based assessment of GTR vs. STR might be suffice instead of Simpson Grade for classifying extent of resection.